# The Influence of Land Use Change on Ecosystem Service Value in Shangzhou District

**DOI:** 10.3390/ijerph16081321

**Published:** 2019-04-12

**Authors:** Keyue Yuan, Fei Li, Haijuan Yang, Yiming Wang

**Affiliations:** 1College of Urban and Environmental Science, Northwest University, Xi’an 710127, China; yuan_keyue@163.com (K.Y.); wymisi@163.com (Y.W.); 2College of Geoscience and Surveying Engineering, China University of Mining & Technology, Beijing 100083, China

**Keywords:** land use change, ecosystem services value (ESV), socio-economic factor correction coefficient, biomass factor correction coefficient China

## Abstract

Land use change has an impact on the ecosystem service value because it changes the structure and function of ecosystems. This paper analyzed the changes in land use during the period from 2000 to 2015 in Shangzhou district, and used the equivalent value of ecological services per unit area of land ecosystem combining the natural and economic conditions of Shangzhou district. Based on this method, the ecological service value of Shangzhou district was estimated, and the impact of land use change on the ecological service value was analyzed. The results showed that: (1) the main types of land use in Shangzhou district were grassland, woodland and farmland, among which the contribution rate of woodland to the value of local ecosystem services was the highest; (2) the overall trend in the ecosystem service value in Shangzhou district increased between 2000 and 2015, from 10.74 × 10^8^ yuan in 2000 to 20.32 × 10^8^ yuan in 2015, which is the result of the combined effects of regional economic development and changes in the natural environment and land use patterns; and (3) the main reason for the value increase of ecosystem services in Shangzhou district between 2000 and 2015 was that the grain-for-green policy transformed a considerable amount of farmland into woodland, while the main reasons for a decline in value was the expansion of built-up land that occupied other types of land.

## 1. Introduction

Ecosystem services refer to the natural conditions and their effectiveness formed by the ecosystem and ecological processes to maintain human survival [1]. The ecosystem supports and maintains the balance of the human living environment by regulating the climate, maintaining biodiversity and so on. It also provides the food and raw materials needed in life and production, and brings entertainment and aesthetic enjoyment to human beings [2]. As Costanza et al. [3] noted in 1997, articles about global ecosystem services value (ESV) assessment, and ecosystem services have become a hot subject of research on ecology and ecological economics. Subsequently, domestic and foreign scholars have studied the evaluation methods proposed by Costanza et al. and have explored and improved the theoretical methods of valuing ecosystem services [4,5,6,7,8,9,10]. On the basis of the research by Costanza et al., the Chinese scholar, Xie Gaodi [11] conducted a questionnaire survey in 2002 among 200 professionals with ecological backgrounds, modified the coefficient according to the actual situation of China, and obtained the land ecosystem service value with China as the scale. This method has the advantages of simple use, less data demand, high comparability of results and comprehensive evaluation, and thus it is used in the study of regional ecosystem service value evaluation in China [12,13,14,15,16]. However, this method provides the unit price of the average ecosystem service value in China, and the ecological service value of the ecosystem is closely related to the region’s natural environment and social and economic conditions. Therefore, the actual situation in the research area should be taken into consideration in the specific study, and the biomass and socio-economic factors related to the national ecosystem service value should be modified so as to obtain the regional ecosystem service value.

Land contains terrestrial ecosystems, and land use is an important activity in the survival and development of human beings. Changes in land use patterns will cause changes in the land cover pattern, which will affect the structure of the ecosystem, thus affecting the service functions and values of the ecosystem [17,18,19]. Therefore, it is very important to understand the impact and interaction of land use change on ecosystem service values, and research on the impact of land use change on regional ecosystem services has been widely implemented in recent years [20,21]. However, earlier studies have mainly focused on ESV change based on land area, and paid more attention to the change of land resource area while ignoring the change in structure and pattern caused by the change of land use area [22,23], especially the change of land use process. For example, land use change in the process of urbanization has a significant impact on the value of ecosystem services [24], and the spatial and temporal differences in ecosystem services value are influenced by regional differences in per capita GDP, population density and urbanization rate [25]. Additionally, the value of land use and regional economic development affect each other and have a well correlated, harmonious relationship [26].

Shangzhou district is located in the south-central part of Shaanxi province and in the southern foothills of the Qinling Mountains, where the ecological environment is excellent. Since the 21st century, the rapid economic development in the region and the acceleration of urbanization have resulted in the continuous expansion of built-up land and the occupation of other types of land. Meanwhile, due to the influence of various factors such as the grain-for-green policy, the type and pattern of land use in the region has undergone great changes, and more importantly, the ecological environment has also been affected. However, a rational land use pattern plays an important role in realizing the coordinated development of a regional economy, society and ecology. This paper analyzes the land use change in Shangzhou district from 2000 to 2015, and discusses the impact of land use change on the ecosystem service value, so as to provide a scientific basis for the government to formulate more rational land use strategies.

## 2. Materials and Methods

### 2.1. Study Area

The Shangzhou district of Shangluo city is located in the south-central part of Shaanxi province and in the hinterland of the southern foot of the Qinling Mountains. The geographical coordinates are between 109°30’–110°14’E and 33°38′–34°12’N (Figure 1). It stretches across the Yangtze River and the Yellow River basins, and is also located at the China north-south natural boundary between the Qinling Mountains and the Huai river. The land is high in the northwest and low in the southeast, with an average elevation of 880 meters. The average annual temperature is 12.8 degrees Celsius, and the average annual rainfall is 740 millimeters. It has a warm temperate zone monsoon and semi-humid mountain climate. Shangzhou district had a total area of 2672 square kilometers with four subdistrict offices, 16 towns and 10 townships. In 2015, the total population of the Shangzhou district was 550,000, of which the agricultural population was about 280,000 and the non-agricultural population was about 270,000, while the regional GDP was about 12,289.9 million yuan. In 2003, when the state implemented the regulations on returning farmland to forest, Shangzhou district started a program to return farmland to forest [27].

### 2.2. Data Sources

The data used in this paper mainly include: (1) Land use data obtained from the database of the Chinese Academy of Sciences Resource and Environmental Science Data Center, including four periods of land use data in Shaanxi province in 2000, 2005, 2010 and 2015; (2) topographic data, GDEMV230M resolution digital elevation data downloaded from the “Geospatial Data Cloud” website; (3) meteorological data downloaded from the “China Meteorological Data Network”, including the national average rainfall and average temperature from 2000 to 2015, etc., and the interpolation model was used to spatially interpolate the station meteorological data; and (4) socio-economic data obtained from the Statistical Yearbook of Shangluo City (2001–2016).

This paper refers to the classification system of Land Resources Classification System of the Chinese Academy of Sciences, the types of land use in the study area were divided into the following six categories: farmland, woodland, grassland, water areas, urban and rural industrial and mining residential land (abbreviated as “built-up land”) and unused land.

The following are available online at http://www.resdc.cn/, land use data including six periods of Shaanxi province in 1980, 1990, 2000, 2005, 2010 and 2015; http://www.gscloud.cn/, GDEMV230M resolution digital elevation data; http://data.cma.cn/, average rainfall and temperature data in Shaanxi from 2000 to 2015.

### 2.3. Calculation of Ecosystem Services Value

In 1997, Costanza et al. proposed a value assessment method for ecosystem services to provide a quantitative assessment of the value of global ecosystem services [3]. Based on the research of Costanza et al., Xie Gaodi et al. [11] divided ecological services into nine categories: gas regulation, climate regulation, water conservation, soil formation and protection, waste treatment, biodiversity maintenance, food production, raw material production, and recreation. Also, they conducted a questionnaire survey of 200 professionals with ecological backgrounds in 2002 and corrected the ecological service value per unit area of China’s terrestrial ecosystem (Table 1) [22]. Estimates of the ecosystem services value were calculated as follows:(1)$$\mathrm{ESV}=\sum A_{k}\times VC_{k}$$

In the formula, ESV is the ecosystem service value (yuan), A_k_ is the distribution area (ha) of the k-th land use type in the study area, and VC_k_ is the ecosystem service value per unit area (yuan/ha) [27]. The water area and the unused land correspond to the ecological system unit ecosystem service value of the water and the desert, respectively. The built-up land had no service value to the ecosystem, so the ecosystem service value per unit area was stipulated as 0.

This method provided the unit price of China’s average ecosystem service value, and the ecosystem service value has a close relationship with its natural environment and socio-economic conditions. Therefore, according to the actual situation in Shangzhou district, the regional ecosystem service value was modified by biomass and socio-economic factors, so as to obtain the ecosystem service value table of Shangzhou district (Table 2 and Table 3). The formula was as follows:(2)$$\mathrm{ESV}=\sum A_{k}\times VC_{k}\times R_{t}\times S_{t}$$

In the formula, R_t_ is the socio-economic factor adjustment coefficient for the t-year of the study area, and *S*_t_ is the biomass factor adjustment coefficient for the t-year of the study area.

Biomass is the quantity of plant-accumulated material per unit area [28]. The larger the total biomass, the stronger the regional ecological service function [29]. This paper used the net primary production potential to modify the biomass and used the Thornthwaite Memorial model [30,31] of actual evapotranspiration to estimate the net primary production potential. The formula was as follows:(3)$$R_{t}=\frac{NPP_{st}}{NPP_{t}}$$
(4)$$\mathrm{NPP}=3000\left[1-e^{-0.0009695\left(V-20\right)}\right]$$
(5)$$V=\frac{1.05R}{\sqrt{1+{\left(1+1.05R/L\right)}^{2}}}$$
(6)$$L=3000+25T+0.05T^{3}$$

In the formula, NPP_st_ is the net primary production potential (t/ha/a) of the average vegetation in Shangzhou district, NPP_t_ is the national average net primary production potential (t/ha/a); V is the annual actual evapotranspiration (mm); L is the annual average evapotranspiration (mm); T is the annual average temperature (°C); and R is the annual precipitation (mm).

In addition, the degree of economic development in Shangzhou district has some differences compared to the whole country so it was necessary to correct the actual socio-economic conditions of Shangzhou district for the willingness to pay and the ability to pay for various services. The formula was as follows:(7)$$S_{t}=P_{t}\times A_{t}$$

In the formula: P_t_ is calculated by the logistic regression model, and it represents peoples’ willingness to pay for ecosystem services; *A*_t_ is calculated from per capita GDP, and it represents peoples’ ability to pay for ecosystem services.
(8)$$A_{t}=\frac{GDP_{st}}{GDP_{t}}$$
(9)$$P_{t}=\frac{W_{st}}{W_{t}}$$
(10)$$W=2/\left(1+e^{-m}\right)$$
(11)$$m=\frac{1}{{\mathrm{En}}_{t}}-2.5$$
(12)$${\mathrm{En}}_{t}={\mathrm{En}}_{tr}\times \left(1-P_{tu}\right)+{\mathrm{En}}_{tu}\times P_{tu}$$

In the formula GDP_st_ is the per capita GDP (yuan/person) in the Shangzhou district; GDP_t_ is the national per capita GDP (yuan/person) in the t year. W_st_ and W_t_ are the willingness of people in the Shangzhou district and whole country, respectively, to pay for the ecosystem services during the t years; En_t_ is the Engel’s coefficient of the study area during the t year; En_tr_ and En_tu_ are the Engel’s coefficient of the town and the rural area, respectively, in the t years; and P_tu_ is the proportion of the urban population in the study area at t year (%).

## 3. Results

### 3.1. Land Use Change

According to the land use data for the four periods of 2000, 2005, 2010 and 2015, the area change was calculated (Table 4), and the results showed that the land use types in Shangzhou district were mainly grassland and woodland, and in 2015, these two areas accounted for 40.92% and 33.83%, respectively, of the total area of the Shangzhou district. Between 2000 and 2015, the area of farmland decreased by 4972.02 ha and that accounted for 7.46% of the total farmland; the area of woodland, grassland, water areas, built-up land and unused land increased, among them, the increase in the area of woodland was the largest. The increase was 2899.93 ha, accounting for 3.35% of the total area; the built-up land had the highest growth rate and increased its area to 1721.79 ha, accounting for 67.56% of the total.

From the change transfer matrix between land use types in Shangzhou district (Table 5), it can be seen that the conversion type was more significant and the conversion area was more than 1000 ha included conversions from farmland to woodland, farmland to grassland, farmland to built-up land, grassland to farmland and grassland to woodland, with conversion areas of 2308.42 ha, 3406.27 ha, 1565.41 ha, 1565.41 ha and 1373.29 ha, respectively.

From the spatial distribution of land use in Shangzhou district (Figure 2), it can be seen that initially the farmland was roughly distributed in Danjiang, Yaoshi, Nanqin, Banqiao and other places, and it was mainly terraces, river valley basins and gentle slope zones. There was also a lot of farmland in the river valley area because the terrain here is flat, and the hydraulic conditions are also good, hence the degree of agricultural intensification degree was high and the distribution of farmland was more concentrated. Woodland and grassland were scattered in mountains and valleys. The water areas mainly include the Danjiang river, Nanqin river, Banqiao river and Dajing river. Built-up land was mainly distributed in towns such as Chengguan. According to the distribution of land use changes in the Shangzhou district from 2000–2015, the most obvious change was the expansion of built-up land around Chengguan. Next, the area of farmland on the steep slopes scattered in beams, ravines, troughs and channels decreased, and it was converted to woodland or grassland. These areas were unsuitable for farming, so they were the key areas for abandoning farmland to woodland or grassland.

### 3.2. Ecosystem Service Value Changes

According to the Chinese ecosystem service value table based on the study of Xie Gaodi et al. [11], and corrected for the natural and socio-economic conditions in Shangzhou district (Table 2 and Table 3), the value map of ecosystem services in Shangzhou district from 2000 to 2015 (Figure 3) was obtained.

Figure 3 shows that the total value of ecosystem services in Shangzhou district increased from 2000 to 2015, increasing from 10.74 × 10^8^ yuan in 2000 to 20.32 × 10^8^ yuan in 2015, of which the value of support services increased by a maximum of 4.99 × 108 yuan to 9.40 × 10^8^ yuan; followed by regulating services, from 4.18 × 10^8^ to 7.94 × 10^8^ yuan; provisioning services increased from 1.17 × 10^8^ yuan to 2.21 × 10^8^ yuan; and cultural services added value increased least from 0.40 × 10^8^ yuan to 0.78 × 10^8^ yuan.

From the perspective of the contribution of different types of land use to the ecosystem service value of Shangzhou district (Figure 4), in 2000, the ecosystem service value of Shangzhou district mainly came from woodland, grassland and farmland, and their contribution to the total value of ecosystem services was 59.73%, 24.69% and 14.53%, respectively. Although the service value per unit area of the water area was high, the proportion of its area to the total area was small, so the contribution rate was only 1.05%. The built-up land had no service value to the ecosystem, so its contribution rate was 0. By 2015, the contribution rate of woodland to the ecosystem service value increased by 1.34% to 61.07%, while the grassland and farmland’s ecosystem service value decreased by 0.21% and 1.23% to 24.48% and 13.31%, respectively. This was due to the implementation of local policies such as the grain-for-green project, which transformed grassland and farmland into woodland, hence the reduction in grassland and farmland area reduced their contribution to the total value of ecosystem services. The contribution of the water area to ecosystem service value increased by 0.10% and reached 1.15%.

Excluding regional economic conditions and the impact of the natural environment, the adoption of the 2015 socio-economic factor correction coefficient and the biomass factor correction coefficient of 1.620 and 0.443 were used to analyze the mechanism of the impact of land use change on the value of local ecosystem services during the study period. From the profit and loss matrix of ecosystem services in the Shangzhou district from 2000 to 2015 (Table 6), under the precondition of harmonizing the social economic factor correction coefficient and the biomass factor correction coefficient, the total value of ecosystem services in Shangzhou district showed a growing trend. The increase was 2373.46 × 10^4^ yuan. In the period from 2000 to 2015, woodland brought the most added value to the ecosystem, with a total growth of 3449.23 × 10^4^ yuan. The increase in water area also increased the ecosystem service value by 345.39 × 10^4^ yuan, and all types of land converted to built-up land ecosystems are reduced. Therefore, the total value of the regional ecosystems caused by the expansion of land for construction are the largest, about 805.33 × 10^4^ yuan; followed by farmland which fell in value by 498.65 × 10^4^ yuan as a result of the decrease in area. There was a slight increase in the grassland area during the study period, but due to the conversion of some woodlands, waters, etc., to grassland, more value was lost, therefore, the grassland value was reduced by 207.65 × 10^4^ yuan. Overall, from 2000 to 2015, the total value of ecosystem services in Shangzhou district increased more than decreased. The value added was mainly due to the increase in the value of grassland and farmland converted into woodland.

From the perspective of the temporal and spatial distribution of ecosystem service value changes in Shangzhou district (Figure 5), the value of ecosystem services increased significantly between 2000 and 2005, and the changes are mainly distributed in Yangxie, Jinlingsi, Yangjinghe, Beikuanping and Yecun and other places, the farmland was converted into woodland or grassland, and ecosystem service value increased mainly in the gully region. The areas where the value of ecosystem services were reduced include the subdistricts of Chenyuan, Chengguan, Dazhaoyu and Shahezi, and the reduction value was 0–5000 yuan/sq. This was due to the expansion of urban residential sites and built-up land to occupy the surrounding farmland and grassland. During the period from 2005 to 2010, the increase in the value of ecosystem services was relatively reduced, and it occurred in areas such as Humuguan and the northwest of Yangxie township. The urbanization rate of Chenyuan, Chengguan, Dazhaoyu and Shahezi town was accelerated, and the built-up land was further expanded and occupied more land of other types. The value of ecosystem services was reduced by 5000–10,000 yuan per square hectare. During the period from 2010 to 2015, the rate of change in the value of ecosystem services slowed down, and there was almost no growth in the value areas. The urbanization of Chengguan, Dazhaoyu and Liuwan developed further while other types of land were converted into built-up land. The most obvious change during this period was a Significant reduction in the ecosystem service value.

## 4. Discussion

Based on the Chinese terrestrial ecosystem service value scale studied by Xie et al., this paper modified the biomass factor and socio-economic factor coefficient of the ecosystem service value scale based on the natural and socio-economic conditions of Shangzhou district, and obtained the ecosystem service value scale of Shangzhou district. This is more suitable for the evaluation of the ecosystem service value in the region, and the results are more accurate and reliable and better reflect the dynamic changes in the ecosystem service value in Shangzhou district.

According to the analysis results, the main types of land use in Shangzhou are grassland, woodland and farmland, which accounted for 40%, 33.83% and 23.31%, respectively, of the total area of Shangzhou district in 2015. From 2000 to 2015, due to the influence of local policies such as returning farmland to forest and grass, a large amount of farmland scattered on the steep slopes in depressions, troughs and channels was converted into woodland and grassland [32]. In addition, with the further development of the local social economy and the intensification of urbanization, the boundaries of the construction area increased [33,34]. The conversion of a large amount of cultivated land to woodland is mainly due to the implementation of the policy of returning farmland to forest, while the conversion of farmland to built-up land is caused by the occupation of cultivated land due to the intensification of urbanization and the expansion of built-up land [35,36,37]. In addition, under the dual effects of the natural environment and social and economic conditions [38,39,40], there has also been transformation between water areas and woodland, grassland and built-up land, as well as mutual transformation between woodland, grassland and built-up land. The value of ecosystem services in Shangzhou district increased from 10.74 × 10^8^ yuan in 2000 to 20.32 × 10^8^ yuan in 2015. This was the result of regional economic development, and changes in the ecological environment and land use patterns. The ecosystem service value of Shangzhou district mainly comes from woodland, grassland and farmland, and although the service value per unit area of the water area is relatively high, it occupies a small proportion of the total area, and its contribution rate is low. Under the premise of the same socio-economic and natural condition correction factors, the total value of ecosystem services in Shangzhou district increased by 2373.46 × 10^4^ yuan. The main sources of value added were farmland and grassland converted into woodland, which has unit land ecosystem services with higher value. During 2000–2005, the value of ecosystem services in Shangzhou district increased significantly and the rate was rapid. The value-added regions were mainly distributed in Yangxie, Jinlingsi and Yangyuhe where the farmland was converted into woodland or grassland. During 2005–2010, the rate of change in the value of ecosystem services slowed down. The expansion of built-up land in Chenyuan, Chengguan, Dazhaoyu and Shahezi caused regional devaluation. The rate of change in ecosystem service value slowed again in 2010–2015, with little value growth. The development of urbanization and expansion of built-up land in Chengguan, Dazhaoyu and the northern part of Liuwan significantly reduced their ecological value.

## 5. Conclusions

This article discusses the impact analysis of changes in the value of regional ecosystem services caused by changes in land use, while the actual ecosystem service value is affected by many factors such as natural conditions or the economic environment, hence the value coefficient changes continuously due to the scarcity of resources and people’s needs in different periods. The method for estimating the value of ecosystem services in this article was based on the research of Costanza et al. [3] and Xie Gaodi et al. [11]. However, due to the complexity, dynamics, and regional differences of actual ecosystems, such as different time periods or woodland of different ages, different ecosystem service values will be produced. Furthermore, the ecosystems in different regions had been altered, and the types of land under human influence did not exactly correspond to the natural ecosystem, so the value of the service value of each land use type was also similarly categorized. The results of the final calculation are only a rough estimate and are for reference only.

Land is the structural basis of terrestrial ecosystems and changes in land use will alter the structure and function of ecosystems, hence they will also affect the service value of regional ecosystems. In addition, land use is one of the important activities in human survival and development and human activities affect the impact of land use and further affect the service value of regional ecosystems. This paper analyzed the changes in land use in Shangzhou district during the period 2000–2015 based on land use data, we used the equivalent value of ecological services per unit area of land ecosystem combining the natural and economic conditions of Shangzhou district with biomass and socio-economic factors correction, we estimated the ecological services value of Shangzhou district and analyzed the impact of land use change on the ecosystem services value. The results showed: (1) the value of ecosystem services in Shangzhou district increased from 2000 to 2015 as a result of regional economic development, ecological environment change and land use change, and (2) from 2000 to 2015, the increase of the ecosystem service value in Shangzhou district was mainly due to the large increase in woodland area caused by the policy of returning farmland to woodland The decrease was mainly due to the occupation of other types of land for urbanization and construction land expansion.

## Figures and Tables

**Figure 1 ijerph-16-01321-f001:**
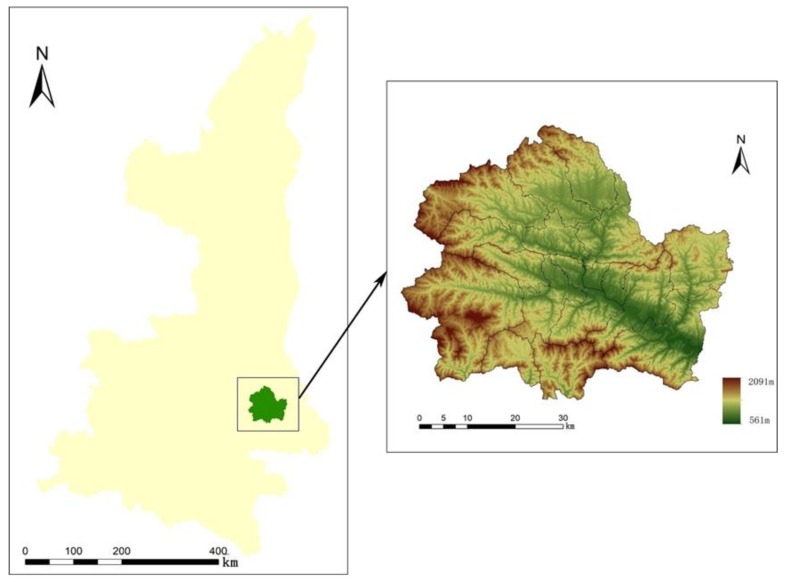
Location of Shangzhou district.

**Figure 2 ijerph-16-01321-f002:**
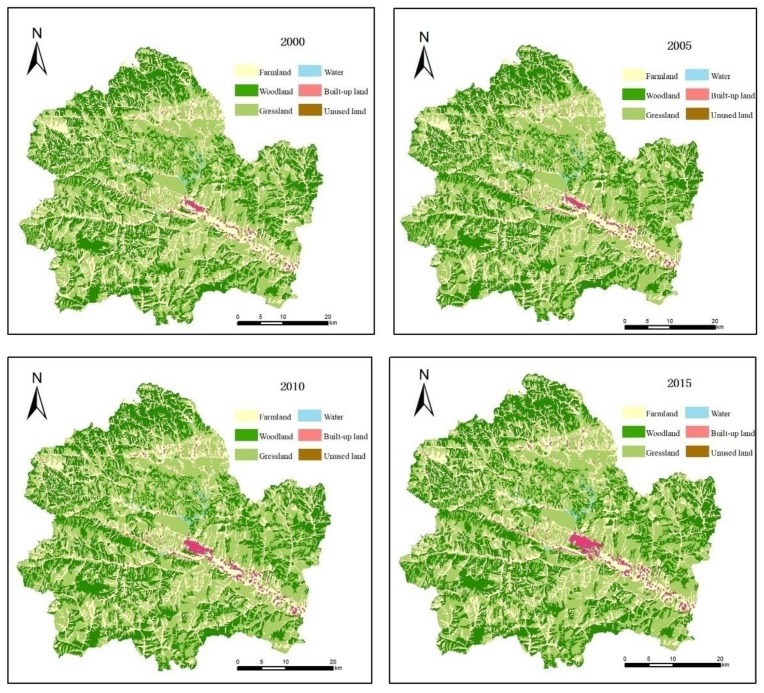
Land use space change in Shangzhou district.

**Figure 3 ijerph-16-01321-f003:**
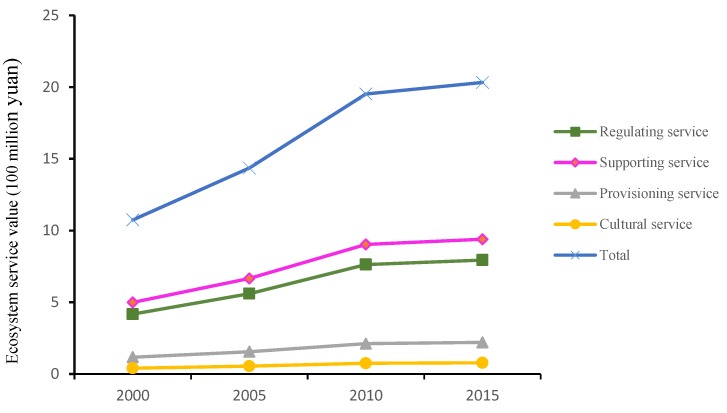
Ecosystem service value of Shangzhou district from 2000 to 2015.

**Figure 4 ijerph-16-01321-f004:**
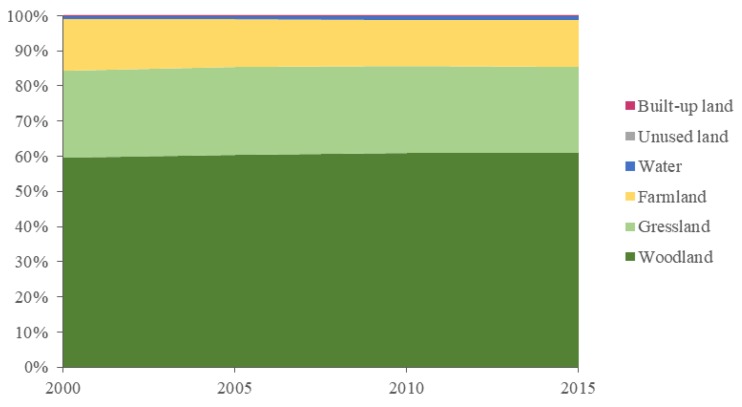
Contribution rate of land use types in Shangzhou district to ecosystem service value between 2000 and 2015.

**Figure 5 ijerph-16-01321-f005:**
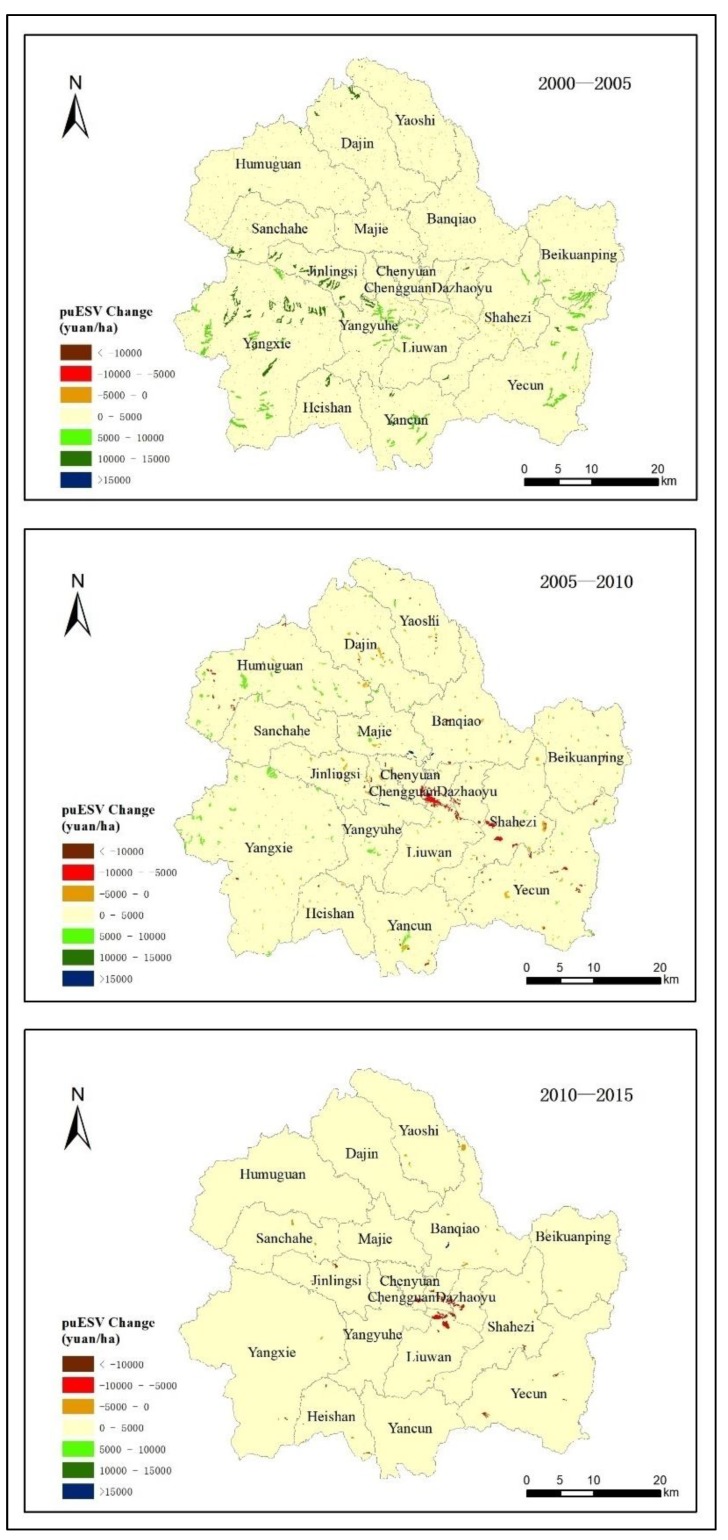
Changes in ecosystem service value in Shangzhou district from 2000 to 2015.

**Table 1 ijerph-16-01321-t001:** Ecosystem services value per unit area of different terrestrial ecosystems in China [22] (yuan/ha/yr).

Land Use	Farmland	Woodland	Grassland	Water	Unused Land	Built-Up Land
Ecosystem	Farmland	Forest	Grassland	Water	Desert	—
Regulating service	1760.8	8317.6	2212.2	18440.2	26.5	0
Supporting service	3371.3	7494.7	3849.2	18298.7	327.3	0
Provisioning service	973.4	2389.1	309.7	97.3	8.8	0
Cultural service	8.8	1132.6	35.4	3840.2	8.8	0
Total	6114.3	19334	6406.5	40676.4	371.4	0

**Table 2 ijerph-16-01321-t002:** Adjustment coefficient of NPP in Shangzhou district from 1980 to 2015.

Years	2000	2005	2010	2015
R_t_	1.764	1.797	1.706	1.620

**Table 3 ijerph-16-01321-t003:** Socio-economic correction coefficient of Shangzhou district from 2000 to 2015.

Years	2000	2005	2010	2015
S_t_	0.217	0.283	0.403	0.443
A_t_	0.215	0.282	0.398	0.439
W_t_	1.008	1.002	1.013	1.007

**Table 4 ijerph-16-01321-t004:** Land use changes in Shangzhou district from 2000 to 2015.

Land Type	2000	2005	2010	2015	Rate of Increase	Annual Rate of Increase
	Percent of Total Area	Percent of Total Area	Percent of Total Area	Percent of Total Area		
Farmland	25.19	23.41	23.30	23.31	−7.46	−0.50
Woodland	32.74	33.40	33.88	33.83	3.35	0.22
Grassland	40.83	41.79	41.09	40.92	0.21	0.01
Water area	0.27	0.27	0.30	0.30	10.47	0.70
Built-up land	0.96	1.12	1.43	1.61	67.56	4.50
Unused land	0.00	0.00	0.01	0.02	614.29	40.95

**Table 5 ijerph-16-01321-t005:** Land use type change transfer matrix in Shangzhou district (ha).

2000	2015						Total
	Farmland	Woodland	Grassland	Water Area	Built-up Land	Unused Land	
Farmland	59311.24	2308.42	3406.27	34.44	1565.41	16.06	66641.84
Woodland	501.59	85813.63	257.56	1.76	19.85	16.99	86611.38
Grassland	1814.26	1373.29	104577.30	62.09	198.93	9.78	108035.65
Water area	0.90	8.69	13.14	703.70	0.00	0.00	726.42
Built-up land	41.74	7.28	13.09	0.08	2486.73	0.21	2549.13
Unused land	0.10	0.00	0.00	0.00	0.00	6.74	6.83
Sum	61669.82	89511.31	108267.35	802.07	4270.92	49.79	264571.26
Area change	−4972.02	2899.93	231.70	75.65	1721.79	42.95	—
Area change ratio	50.00	29.16	2.33	0.76	17.31	0.43	100.00

**Table 6 ijerph-16-01321-t006:** Profit and loss of ecosystem service value in Shangzhou district from 2000 to 2015 (10^4^ yuan).

2000	2015						Total
	Farmland	Woodland	Grassland	Water Area	Built−Up Land	Unused Land	
Farmland	0.00	2188.49	71.38	121.93	−686.41	39.80	1735.20
Woodland	−475.53	0.00	−238.78	4.57	−27.53	26.01	−711.25
Grassland	−38.02	1273.16	0.00	218.57	−91.40	24.05	1386.36
Water area	−3.17	−22.52	−46.26	0.00	0.00	0.00	−71.95
Built−up land	18.30	10.10	6.01	0.31	0.00	0.62	35.34
Unused land	−0.24	0.00	0.00	0.00	0.00	0.00	−0.24
Total	−498.65	3449.23	−207.65	345.39	−805.33	90.48	2373.46

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
