# Peer review of "The Influence of Land Use Change on Ecosystem Service Value in Shangzhou District"

_ijerph, 2019, doi:10.3390/ijerph16081321_

Reviewer 1 Report

The paper shows the land use changes in years 2000-2015 and influence these changes on ecosystem service value in Shangzhou district. The paper structure is good and methodology is correct. Below I mention some problems with your article:

1.  Introduction:

1.1. Please indicate the main aims of this article related to its title.

1.2.  Please give same information about the category of ecosystem service and related references.

2. Materials and Methods section needs more information about study area, e.g. population changes and dynamic, sector structure of GDP and about implementation of green policy (local policies) and projects.

3.  Please compare the results presented in the Discussion paragraph from others regions in China.

 Specific comments were provided directly on the pdf version (enclosed pdf).

Author Response

Point 1: Indicate the main aims of this article related to its title.

Response 1: Changes in land use change the structure and function of the ecosystem, thus affecting the service value of the regional ecosystem, and ultimately affecting the ecosystem and the quality of people's daily living environment. Therefore, rational land use pattern plays an important role in realizing the coordinated development of regional economy, society and ecology. Based on this consideration, many experts and scholars began to study land use, focusing on the value of ecosystem services and taking it as a comprehensive indicator of the ecological benefits of land use change.

According to the culture area of land use data (2000-2015), this paper analysis the change of land use within the research phase, and based on the terrestrial ecosystem per unit area ecosystem service value, combining with the culture area of natural and social economic situation, estimate culture area ecosystem service value and discuss the land use change on the influence of regional ecosystem services value, in order to for the government to provide scientific evidence for a more reasonable land use strategies.

And we have supplemented and improved the main aims in the introduction of the revised manuscript on page 2, see LOA2.

Point 2:Provide same information about the category of ecosystem service and related references.

Response 2: Thank you for pointing this out, We have added the information about the category of ecosystem service and related references in page 3, see LOA3.

Point 3: Materials and Methods section needs more information about study area, e.g. population changes and dynamic, sector structure of GDP and about implementation of green policy (local policies) and projects.

Response 3: Thank you for pointing this out, We have add more information about study area, see LOA4.

Point 4: Compare the results presented in the Discussion paragraph from others regions in China.

Response 4: Thank you very much for your suggestion, but first of all, this paper calculates the value of ecosystem services based on the situation of Shangzhou district, and it aims to provide reference for the local government to formulate appropriate land policies. Secondly, due to the time, the value of ecosystem services in Shangzhou district was not compared with other regions in the same period. However, I think this is a very meaningful research direction, which can be further studied later.

Point 5: Other comments and amendments in PDF.

Response 5: The other revisions noted in the PDF are directly in the revised manuscript.

Reviewer 2 Report

Dear authors,

I am very sorry to write this, but your paper (in its current form) does not deserve publication. To be honest, I don't think that this can be changed, as I consider the very approach of your study seriously flawed. You are trying to use quite outdated and methodologically problematic values from the Costanza et al (1997) study and transform it in dubious and extremely rough ways to fit your case study region, to then calculate per-land-use-category aggregate values... Given the source of the data, the way they were transformed, they are not much more than guesstimates. Their informational value is close to zero. In addition, you do not report essential information about the data you used and the methodological approaches for the various transformation steps. Last but not least, the English of the paper is very bad and quite difficult to understand. Given all this, I have to recommend rejection of your paper.

P.S. I upload a commented version of the manuscript PDF. I only read through the end of section 2 - at this point it had become sufficiently clear that this paper has no future.

Author Response

Point 1: I am very sorry to write this, but your paper (in its current form) does not deserve publication. To be honest, I don't think that this can be changed, as I consider the very approach of your study seriously flawed. You are trying to use quite outdated and methodologically problematic values from the Costanza et al (1997) study and transform it in dubious and extremely rough ways to fit your case study region, to then calculate per-land-use-category aggregate values... Given the source of the data, the way they were transformed, they are not much more than guesstimates. Their informational value is close to zero.

Response 1: Thank you for pointing this out. In the evaluation of ecological benefits, the international practice is to integrate studies in different regions and summarize the value of main ecological process functions and ecosystem benefits through statistics. Costanza et al. published "The value of the word's ecosystem services and natural capital" in the Nature at 1997, the principles and methods of ecosystem service value estimation are clarified in a scientific sense. I understand that some data in this study are quite different, so this study has also been severely criticized. However, the study of Xie Gaodi et al. aims at the shortcomings of Costanza et al., and at the same time refers to some reliable results. Based on the questionnaire survey of 200 Chinese ecologists, the equivalent factor table of the value of ecosystem ecosystem services in China is developed, which, in my opinion, has certain reference value. This paper based on the terrestrial ecosystem per unit area ecosystem service value, combining with the culture area of natural and social economic situation, estimate culture area ecosystem service value and discuss the land use change on the influence of regional ecosystem services value, so as to provide a scientific basis for the government to formulate more reasonable land use strategies.

Point 2: In addition, you do not report essential information about the data you used and the methodological approaches for the various transformation steps.

Response 2: Thank you for your comments. Due to the limitation of the article length, the article only includes the basic data sources, formulas and calculated natural and economic adjustment coefficients, and also lists the revised ecosystem services value per unit area of different terrestrial ecosystems in Shangzhou district.

Point 3: Last but not least, the English of the paper is very bad and quite difficult to understand. Given all this, I have to recommend rejection of your paper.

Response 3: Regarding the questions you pointed out about the English language in our article,the revised manuscript has been edited for English usage, syntax, word choice and sentence structure, see LOA1.

Point 4: I upload a commented version of the manuscript PDF. I only read through the end of section 2 - at this point it had become sufficiently clear that this paper has no future.

Response 4: Thank you for your valuable suggestions for our article. As for the questions raised in the manuscript PDF, we have made modifications in the manuscript without affecting the results.

Reviewer 3 Report

The authors explore the impacts that land use changes in the Shangzhou district of China have had on ecosystem services per unit area from 2000 to 2015 using values derived from socio-economic importance and willingness to pay, and biomass/productivity. They found that woodland made the highest contributions to ecosystem services, which led to an estimated near-doubling of ecosystem service value over the study period due largely to policies encouraging the transformation of farmland into woodland. Conversely, the main reason for value decline was urban expansion.

Introduction, methods and results all present their content clearly and read well.

The paper treats all land use/land cover types as effectively uniform regardless of age or any deeper consideration of quality. There is nothing inherently wrong with this and it is made clear in the methods that each type of land use has a given value per unit area, and that these values are based on previous published papers, but I think there could be room for some interesting discussion points around the notion that not all woodland, for example, is equal. One of the assumptions of the research, by my understanding, is that any ‘new’ woodland areas in 2015 are assumed to have the same ecosystem value as woodland areas in 2000. However, by some metrics it may be the case that older, more developed woodlands provide ecosystem service values that young woodlands less than 15 years old may not. I fully appreciate that such investigation is beyond the scope of the research presented here, but this issue of woodland age and quality and the potential uncertainty around it may be worth bringing up in the Discussion as a point for future research.

The Conclusion is effective at highlighting first the main contributions and findings of this paper, and second the limitations of the research. If the two paragraphs were switched in order, however, it might help the paper have more impact - certainly do present the limitations, the complexities that make it difficult to apply the results here to other locations, but then allow the paper to finish on a ‘high note’ of what positive contributions have been discovered by this work and how they CAN be usefully applied to research in other places.

There are some minor grammatical/spelling errors (e.g. L28, ‘it also providing…’; L83 ‘there gional’ should be ‘the regional’) but for the most part the paper reads clearly and flows well.

Author Response

Point 1: The paper treats all land use/land cover types as effectively uniform regardless of age or any deeper consideration of quality. There is nothing inherently wrong with this and it is made clear in the methods that each type of land use has a given value per unit area, and that these values are based on previous published papers, but I think there could be room for some interesting discussion points around the notion that not all woodland, for example, is equal. One of the assumptions of the research, by my understanding, is that any ‘new’ woodland areas in 2015 are assumed to have the same ecosystem value as woodland areas in 2000. However, by some metrics it may be the case that older, more developed woodlands provide ecosystem service values that young woodlands less than 15 years old may not. I fully appreciate that such investigation is beyond the scope of the research presented here, but this issue of woodland age and quality and the potential uncertainty around it may be worth bringing up in the Discussion as a point for future research.

Response 1: Thank you for your suggestions. We also agree that the age and quality of woodlands may have an uncertain impact on the ecosystem services value they provide, and not only on woodlands, but also on other aspects. Thank you very much for understanding that such a survey is beyond the scope of this study and is not the focus of this study. We are only concerned with the impact of changes in land use over different time periods on the ecosystem services value. In the last paragraph of the discussion part of the article, this issue has been raised as a direction of future research, see LOA6.

Point 2: The Conclusion is effective at highlighting first the main contributions and findings of this paper, and second the limitations of the research. If the two paragraphs were switched in order, however, it might help the paper have more impact - certainly do present the limitations, the complexities that make it difficult to apply the results here to other locations, but then allow the paper to finish on a ‘high note’ of what positive contributions have been discovered by this work and how they CAN be usefully applied to research in other places.

Response 2: Thank you for the suggestion that you have pointed out. The revised manuscript has adjusted the Conclusions to a more appropriate order, and has reworked the positive contributions found in this work and how they can be effectively applied to research in other places, see LOA7.

Point 3: There are some minor grammatical/spelling errors (e.g. L28, ‘it also providing…’; L83 ‘there gional’ should be ‘the regional’).

Response 3: Thank you for pointing out these grammatical and spelling errors, we have made corresponding modifications in the new manuscript, see LOA1.

Reviewer 4 Report

Please explain why 2000 to 2015? L8 to L13: please rewrite L87: why not used RS data to generate landuse map Fig1: show china at first L151: water area being increased. How? Table 4: justify to one page Table 5: show trend line, figure Fig2: add a bar plot also Fig4: is unclear Table 6: change it to figure

Author Response

Point 1: Please explain why 2000 to 2015?

Response 1: Regarding the selection of the study period, this is because the Chinese government officially implemented the “Returning Farmland to Forests Regulations” in January 2003. Returning farmland to forests is to protect and improve the ecological environment. The sloping land that is easy to cause soil erosion is planned, and the cultivation is stopped in a step-by-step manner. According to the principle of suitable land, trees are planted in accordance with local conditions, and forest vegetation is restored. The construction of returning farmland to forest project includes two aspects: one is to return farmland to forest on slope farmland; the other is to afforestation in barren hills and wasteland.

Point 2: L151: water area being increased. How?

Response 2: Thank you for pointing this out. The changes in the land types of different types in this paper are based on the plaque data on the land use type maps in different periods. However, the reasons for the increase in specific waters are beyond the capabilities of this paper and are not the focus of this paper. However, in the conclusion part of this paper, it illustrates some shortcomings and problems that remain unsolved in this paper.

Point 3: L8 to L13: please rewrite; Table 4: justify to one page; why not used RS data to generate land use map Fig1, show china at first; Table 5: show trend line; Fig2: add a bar plot also; Fig4: is unclear; Table 6: change it to figure

Response 3: Thank you for pointing these out, without affecting the results, we have made modifications in the revised manuscript where possible.